# Analytical code sharing practices in biomedical research



Nitesh Kumar Sharma[1,*], Ram Ayyala[2,*], Dhrithi Deshpande[1], Yesha Patel[1], Viorel Munteanu[3], Dumitru Ciorba[3], Viorel Bostan[3], Andrada Fiscutean[4], Mohammad Vahed[1], Aditya Sarkar[5], Ruiwei Guo[6], Andrew Moore[7], Nicholas Darci-Maher[8], Nicole Nogoy[9], Malak Abedalthagafi[10,11] and Serghei Mangul[1,2]

[1] Titus Family Department of Clinical Pharmacy, University of Southern California, Los Angeles, California, United States
[2] Quantitative and Computational Biology Department, University of Southern California, Los Angeles, California, United States
[3] Department of Computers, Informatics and Microelectronics, Technical University of Moldova, Chisinau, Moldova
[4] Faculty of Journalism and Communication Studies, University of Bucharest, Bucharest, Romania
[5] School of Computing and Electrical Engineering, Indian Institute of Technology Mandi, Kamand, Himachal Pradesh, India
[6] Department of Pharmacology and Pharmaceutical Sciences, University of Southern California, Los Angeles, California, United States
[7] Daniel J. Epstein Department of Industrial and Systems Engineering, University of Southern California, Los Angeles, California, United States
[8] Computational and Systems Biology, University of California, Los Angeles, Los Angeles, California, United States
[9] GigaScience Press, Shek Mun, Hong Kong
[10] Department of Pathology & Laboratory Medicine, Emory University Hospital, Atlanta, Georgia, United States
[11] King Salman Center for Disability Research, Riyadh, Saudi Arabia
* These authors contributed equally to this work.

Corresponding authors
Ram Ayyala, rayyala@usc.edu
Serghei Mangul, mangul@usc.edu

## ABSTRACT

Data-driven computational analysis is becoming increasingly important in biomedical research, as the amount of data being generated continues to grow. However, the lack of practices of sharing research outputs, such as data, source code and methods, affects transparency and reproducibility of studies, which are critical to the advancement of science. Many published studies are not reproducible due to insufficient documentation, code, and data being shared. We conducted a comprehensive analysis of 453 manuscripts published between 2016–2021 and found that 50.1% of them fail to share the analytical code. Even among those that did disclose their code, a vast majority failed to offer additional research outputs, such as data. Furthermore, only one in ten articles organized their code in a structured and reproducible manner. We discovered a significant association between the presence of code availability statements and increased code availability. Additionally, a greater proportion of studies conducting secondary analyses were inclined to share their code compared to those conducting primary analyses. In light of our findings, we propose raising awareness of code sharing practices and taking immediate steps to enhance code availability to improve reproducibility in biomedical research. By increasing transparency and reproducibility, we can promote scientific rigor, encourage collaboration, and accelerate scientific discoveries. We must prioritize open science practices, including sharing code, data, and other research products, to

ensure that biomedical research can be replicated and built upon by others in the scientific community.

## INTRODUCTION

Modern biomedical research relies heavily on quantitative analysis, necessitating mechanisms that ensure transparency, rigor, and reproducibility. Since 2011, an alarming number of retracted articles have been quantified in a "retraction index" indicating a strong correlation with journal impact factor (*Fang & Casadevall, 2011*). This "retraction index" highlighted weaknesses in the publishing system (*Van Noorden, 2011*), such as scientific misconduct, fraud or fabrication, and irreproducible results (due to lack of data or source code) further fueling a lack of trust in science. A total of 12 years after the "retraction index", the number of articles being retracted has increased, from 45 per month in 2010 to 300 retractions per month in 2022, as reported by Retraction Watch, many of which are still due to data fabrication and irreproducible results. Another problem of the rise in the number of retractions is the overwhelming number of article mills and fake articles (*Oransky, 2022*). Typically, retractions are most often prompted by errors, plagiarism, duplicate publication, fraud or suspected fraud, and invalid peer review. These retracted articles are generally associated with journals boasting low impact factors (*Wang et al., 2019*). Researchers play a crucial role in alleviating such issues by prioritizing transparency and reproducibility in their work, thereby fostering increased trust within the scientific community.

While transparency and reproducibility are vital, achieving them necessitates the sharing of well-documented analytical code and data. However, analytical code sharing is not enforced as strictly as data sharing ("Code share," *Nature Editorial, 2014*; "Rebooting review," *Nature Biotechnology Editorial, 2015*), and the effectiveness of existing initiatives is uncertain. Well-structured data and code not only enhance reproducibility, but also provide numerous benefits to researchers–they facilitate swift installation and rerun of analyses, enable visualization, and make analyses more robust against code errors and other mistakes (*Cadwallader et al., 2022*; *Baker et al., 2014*). By contrast, the limited availability of code and associated research products poses challenges to reproducibility across scientific disciplines (*Serghiou et al., 2021*; *Baker, 2016*; *Hutson, 2018*; *Barnes, 2010*). While funding agencies and peer-reviewed journals have recommended best practices, guidelines, and checklists for researchers, including for sharing of analytical code (*Serghiou et al., 2021*; *Fanelli, 2018*), these recommendations often have little impact on the actual availability of research products accompanying research studies (*Serghiou et al., 2021*). This situation is present despite the rise of data-driven biomedical research, and the growing recognition within the broader scientific community about the significance of sharing code used for analysis (*Gomes Dylan et al., 2022*).

In recent years, there has been a growing interest in promoting open science practices in biomedical research. This study aims to explore the patterns and practices of code sharing within this context. To accomplish this, we conducted a comprehensive analysis of 453 manuscripts published in eight scientific journals between 2016 and 2021. The focus of our investigation was to understand the extent to which analytical code, a crucial component of research, is being made available to the broader scientific community. By shedding light on the current state of code sharing, our research seeks to address the need for enhanced reproducibility and collaboration in biomedical research. We propose raising awareness of code sharing practices and taking immediate steps to enhance code availability to improve reproducibility in biomedical research. By increasing transparency and reproducibility, we can promote scientific rigor, encourage collaboration, and accelerate scientific discoveries. We must prioritize open science practices, including sharing code, data, and other research products, to ensure that biomedical research can be replicated and built upon by others in the scientific community. Portions of this text were previously published as part of a preprint (https://www.biorxiv.org/content/10.1101/2023.07.31.551384v3.full).

## MATERIALS AND METHODS

### Download and prepare scientific publications for analysis

We downloaded the full text of 12,603 publications between the years 2016 and 2021 from the PubMed Central (PMC) open access *corpus* in Extensible Markup Language (XML) format from eight journals: Nature Biotechnology, Genome Medicine, Nature Methods, Genome Biology, Bioinformatics, BMC Bioinformatics, Nucleic Acid Research, and Nature Genetics, as these are the leading journals in computational biology and bioinformatics as they are well respected and have a high impact factor. The 12,603 articles we downloaded were all of the articles that were available on the PMC Open Access *Corpus* Database for those eight journals at the time of downloading. We included the commercial use and non-commercial use subsets. From these 12,603 manuscripts, 10 manuscripts were randomly selected from each journal for each year between 2016–2021, totaling to 480 randomly selected articles. Each publication had a unique PMC identification number (PMC ID). The XML file for each publication was stored in the corresponding directory based on the journal the publication came from. To further evaluate these manuscripts for sharing the code and data, we filtered out 27 manuscripts which were published as research letters, reports or editorial *etc*. In total, we had 453 manuscripts available for downstream analysis (Fig. S1).

### Extract publication dates of selected publications

We created a key file with the paths to every XML file representing a publication. We then used a Python script utilizing the package ElementTree to systematically extract publication metadata from these XML files. To find the date the publication was published, our script considered each publication date that was listed in a publication's file (there were often multiple). When the year was missing from a date, we disregarded the date completely. When the month was missing, we recorded the publication month as December. When the day was missing, we recorded the publication day as the last day of

the publication month. After completing this process with each listed date, the earliest converted date was recorded and associated with our record of the publication.

## Analysis of extracted information

We also recorded various parameters such as the raw data availability, where the data is shared, the analytical code availability, code availability in Supplemental Files, code availability in repositories or webpages, presence of code and data availability statements, and the number of citations for each study. All parameters recorded, using our python script, were based on information provided by the authors in the publication. All code and data sharing links were double checked manually to evaluate if links were working. Amongst the 480 studies, we filtered out the studies that were conducting secondary analyses while looking into raw data availability. When we looked into the raw data availability in each article, we assigned 'Yes' and 'No' to indicate availability and unavailability of raw data. The data was considered available if the data and its corresponding dependencies could be stored on public repository operating systems, and the data can produce expected results from the input data with no errors. All data considered in this analysis was real data, meaning no synthetic data available in any of the articles analyzed as we recorded whether the data was shared on 'Gene Expression Omnibus (GEO)', 'Sequence Reads Archive (SRA)' or other repositories such as Bioproject, ENCyclopedia Of DNA Element (ENCODE), DNA Database of Japan (DDBJ), GitHub, ArrayExpress, European Nucleotide Archive (ENA). Those articles which shared data on either DDBJ or ENA were included under 'SRA'. Some articles had data shared or obtained from multiple datasets for different parts of their study and these articles were included under the category 'Multiple'. The category 'Multiple' not only refers to articles which have split their data on multiple repositories but also could refer to those which have duplicated datasets on different repositories.

Similarly, we recorded the code availability in all the 480 articles and assigned 'Yes' or 'No' to indicate availability and unavailability of analytical code. The goal of this study was not to check the completeness of analytical code but the presence and incentive of researchers to share code. We specifically looked for code used for analysis (not to be confused with software or source code for algorithms and tools) and considered code availability as a 'Yes' if any analytical code was shared. For those articles that shared code, we recorded if the code was hosted on GitHub, shared in the Supplemental Materials or hosted other repositories such as Zenodo or a webpage. Some articles that shared code in both Supplemental Materials and on GitHub were included under GitHub. Verifying the completeness and/or accuracy of the code shared was beyond the scope of this study. Additionally, we also recorded whether the articles had data availability statements and code availability statements mentioned in a separate section for easy accessibility. We assigned a 'No' if the article did not have a statement but mentioned the availability of data or code amidst the main text. We also recorded the number of citations the articles had on Google Scholar as of 15 November, 2021, and the policies of the ten journals on sharing code and data. We classified the code and data sharing policies of the journals as

**Table 1 Overview of code and data sharing policies across eight biomedical journals.**

| Journal | Code policy | Data policy |
|---|---|---|
| Bioinformatics | Mandatory | Mandatory |
| BMC Bioinformatics | Encouraged | Encouraged |
| Genome Biology | Mandatory | Mandatory |
| Genome Medicine | Encouraged | Encouraged |
| Nature Biotechnology | Mandatory | Mandatory |
| Nature Genetics | Mandatory | Mandatory |
| Nature Methods | Mandatory | Mandatory |
| Nucleic Acids Research | Encouraged | Mandatory |

**Note:**
Each row documents the journal name, along with its corresponding code and data policies.

'Mandatory', 'Encouraged/Mandatory' if it was unclear, and 'No policy' for those that did not mention any guidelines for sharing code or data.

## Statistical analysis

Chi-square tests of independence were used to identify significant differences between groups in most of the analyses. To specifically determine whether journal policies had an effect on code and data availability, the standard odds ratio was calculated and a Fisher's exact test was conducted to determine whether the odds ratio was significant. All statistical tests were carried out using scipy 1.11.1 package.

## Data availability

All manuscript data discussed in this study is freely available at https://github.com/Mangul-Lab-USC/code-availability. This data was used to produce all results and figures depicted in this study.

## Code availability

All code required to produce figures and analysis performed in this study is freely available at https://github.com/Mangul-Lab-USC/code-availability. The code is distributed under the terms of the General Public License version 3.0 (GPLv3). Source data are provided with this study as described in the Data Availability section.

## RESULTS

### The majority of biomedical studies has limited code availability

In 2015, a study investigating the repeatability in computer science was carried out to see the extent to which computer science researchers share their source code (*Collberg & Proebsting, 2016*). A total of 8 years later, we conducted an extensive study to assess the availability of code and data in biomedical research, analyzing a random sample of 480 articles across eight biomedical journals published from 2016 to 2021 (Table 1). Ten manuscripts were selected from each journal for each year of the study period. Out of 480 manuscripts, we considered 453 manuscripts for further analysis. The filtered

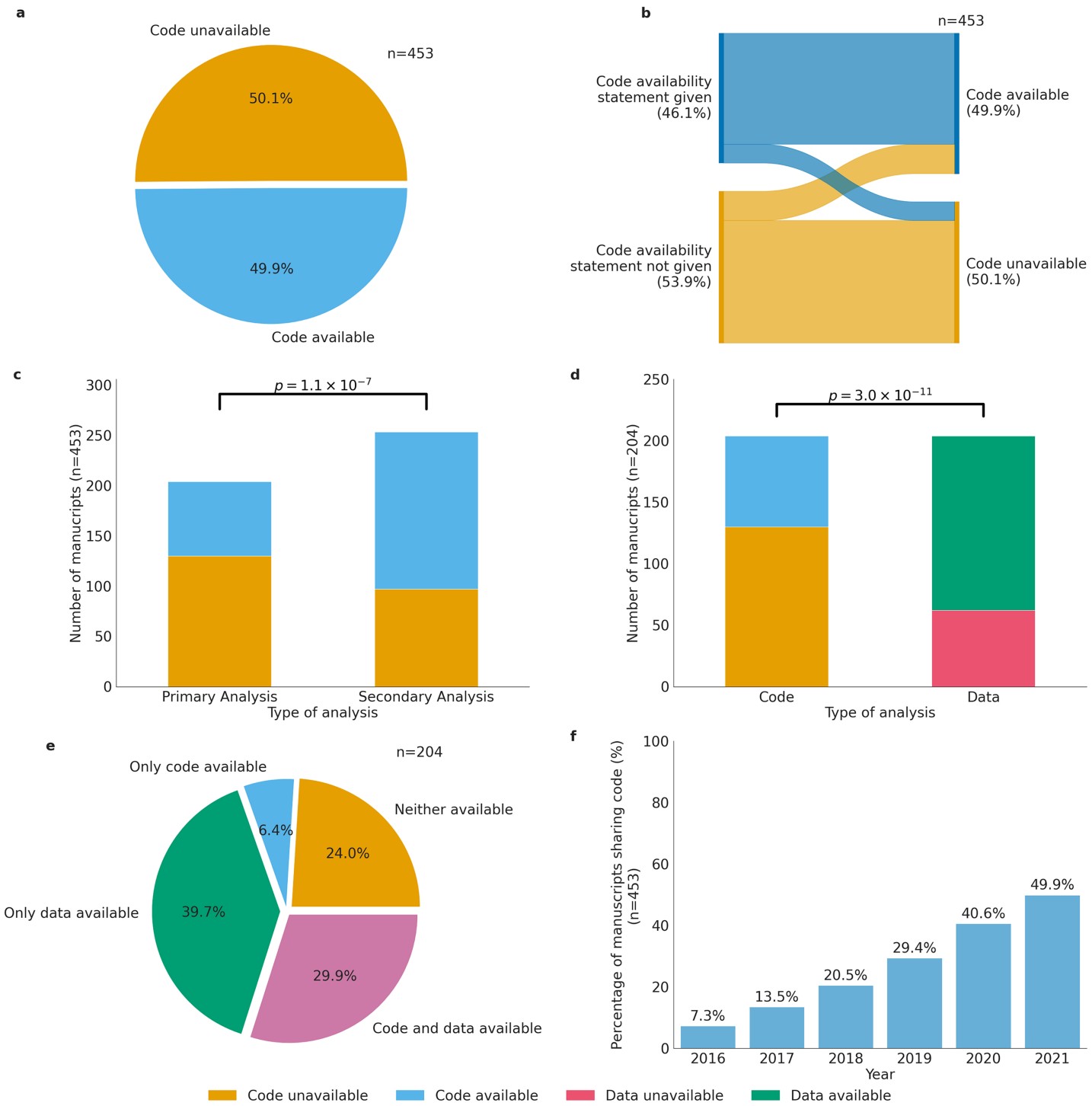

**Figure 1 Code availability in biomedical research.** (A) Availability of code across the 453 biomedical articles. (B) Code availability status across 453 manuscripts on the basis of code availability statement. (C) Number of studies sharing code across primary and secondary analysis studies. Chi-squared statistic: 28.106. $p$-value: $1.15 \times 10^{-07}$ (D) Number of studies sharing code and data across all studies. Chi-squared statistic: 44.16. $p$-value: $3.02 \times 10^{-11}$ (E) Code and data sharing status across primary studies. (F) Cumulative percentage of articles published each year that share code, relative to the total number of articles published from 2016 to 2021 ($n = 453$).

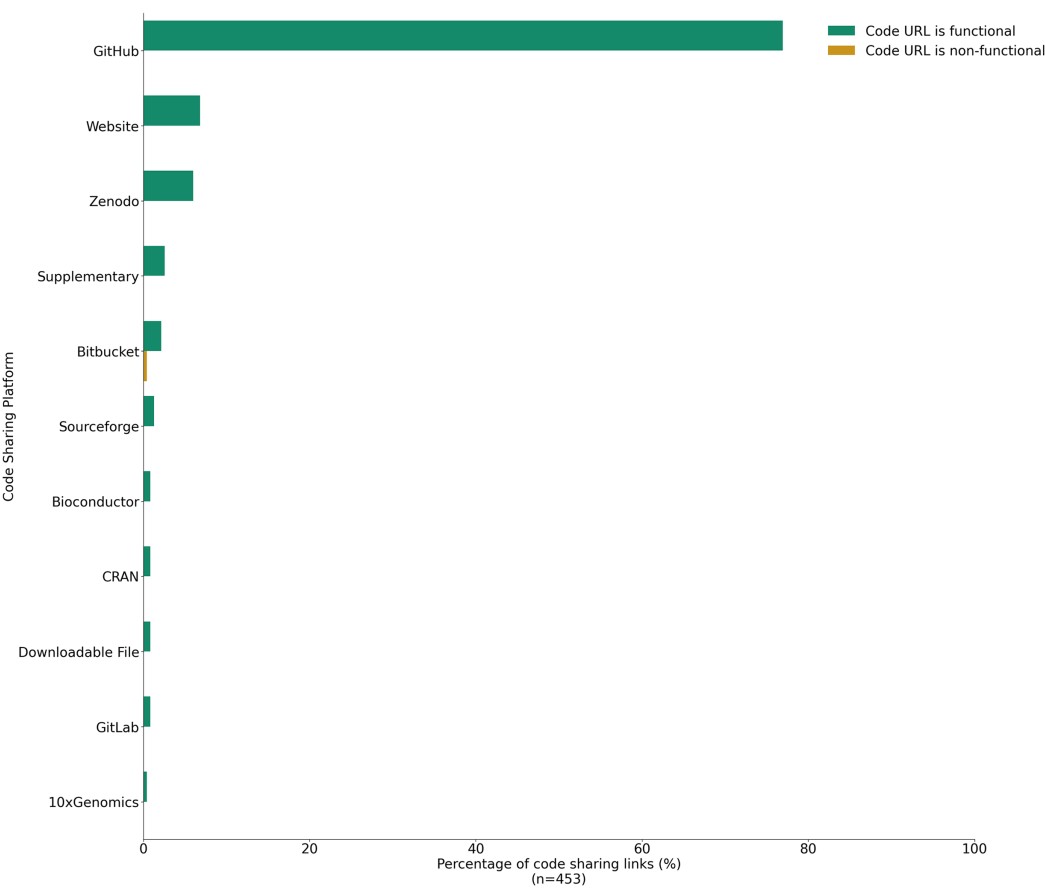

**Figure 2** Percentage of code sharing URLs across all articles from 2016–2021 across eight journals whose links were functional and non-functional ($n = 453$). The data was ordered by first sorting entries where the code sharing links were not expired, followed by ordering these entries in descending order of their frequency counts.

27 manuscripts were research letters, editorials, or research reports which did not report any data or code and were discarded from the analysis. In our analysis, we recorded code availability, data availability in the manuscripts, and the presence of data and code availability statements and code organization. Our results indicate that nearly half (49.9%) of the studies examined failed to share the analytical code used to generate the figures and results used by the researchers (Fig. 1A). Of the code that was shared, the majority (76.9%) was found on GitHub, while 13.7% was hosted on other platforms such as Bitbucket, Sourceforge, or individual websites, 5.9% on Zenodo, 2.6% was included as Supplemental Material, and 0.8% on GitLab (Table S1). Of the 453 studies, 209 (46.1%) used a code availability statement, while 179 (85.6%) of these 209 actually shared their code (Fig. 1B). However, even if the article did share the analytical code, there was still the question of whether the link for the code itself functioned or if it had an archival stable link. The term "archival stable link" refers to a permanent or persistent link that remains functional and accessible over time, ensuring that it continues to point to the same resource or information even if the website structure or content undergoes changes. To assess the archival stability of code sharing repositories, we examined the links of the manuscripts

that did share the code. Surprisingly, we found that less than 1% of the links were unstable, indicating potential issues with long-term accessibility (Fig. 2).

The presence of code availability statements was significantly associated with the availability of shared code (Pearson's $\chi2$ = 42.8, $p$ = $2.7 \times 10^{-9}$). Of the 453 studies analyzed, 49.9% shared code, with 79.2% of them having a code availability statement alongside the shared code, while 20.8% of articles shared code but lacked a code availability statement. In contrast, 43.5% of manuscripts shared neither code availability statements nor code, and 6.6% of manuscripts had a code availability statement but did not share any code (Fig. 1B). To clarify, articles that did not share code typically either did not share any method to obtain their code used in the analysis or the links for the code were broken. The majority of the manuscripts that provided code availability statements included them in the main text of the manuscript along with the data availability statement, or in the results, or the supplementary section.

## Studies performing secondary analysis were more likely to share the code compared to primary analysis studies

We then separated the articles into studies that generated their own data or primary analyses and studies that relied on already available data or secondary analyses. Our analysis revealed that out of the 204 biomedical research articles that conducted primary analyses, 69.6% shared the raw data (Fig. S2). Moreover, we found that 36% of the 204 primary analyses had code and data availability statements, while 44.6% had only given a data availability statement and 0.9% only a code availability statement (Fig. S2). We then investigated the relationship between code availability and the analysis type (primary and secondary analyses) among the 453 articles. We found that 61.7% of the secondary analyses articles shared their code while only 36.3% of primary analyses articles shared their code (Pearson's $\chi2$ = 28.1, $p$ = $1.1 \times 10^{-7}$) (Fig. 1C).

## Data sharing is a more prevalent practice than code sharing in biomedical research

In this study, we investigated the utilization of data repositories and code repositories to assess the prevalence of code and data sharing practices. Of the 44.6% of biomedical research articles that conducted primary analyses between 2016 and 2021, we found that a significant majority had shared data while not sharing their code (Pearson's $\chi2$ = 44.2, $p$ = $3.0 \times 10^{-11}$) (Fig. 1D). Specifically, only 29.9% of the articles were accompanied by both code and data. In contrast, 24.0% of articles shared neither the code nor the data. Of the 204 primary analysis manuscripts, 39.7% were accompanied by data only, whereas 6.4% shared code only (Fig. 1E).

When we examined the cumulative percentage of articles published each year that share code, relative to the total number of articles published from 2016 to 2021 ($n$ = 453), we found that code sharing practices have exhibited a consistent positive trend from 2016 to 2021 with an increase from 7.3% in 2016 to 49.9% in 2021 (Fig. 1F). Data availability appears to be more common than code availability in the field of biomedical research, specifically when looking at primary analysis studies (Fig. S4). When we looked at primary

analyses data year by year, we found that both code and data sharing practices have exhibited a consistent positive trend from 2016 to 2021. Notably, data sharing practices experienced a substantial increase from 9.3% in 2016 to 69.6% in 2021 (Fig. S5), while code sharing practices also demonstrated a noticeable but significantly slower upward trajectory from 1.9% in 2016 to 16.3% in 2021 (Fig. S6). In terms of secondary analysis studies, we found that there is a similar upward trend with an increase from 5.3% in 2016 to 33.5% in 2021 (Fig. S7). In addition to examining the prevalence of data and code availability practices, we also made specific observations regarding the utilization of data repositories.

Our analysis revealed that GEO emerged as the most commonly used repository for sharing data used in analyses, comprising 28.8% of the shared data, followed by SRA at 17.8%. The remaining 47.26% of shared data originated from other repositories (Fig. S8). Additionally, 65.2% of the articles that shared raw data had a data availability statement mentioned in the manuscript (Fig. S9). To explore the relationship between the presence of data availability statements and data sharing, we conducted an analysis and found a statistically significant association between the two (Pearson's $\chi2 = 125.1$, $p = 6.1 \times 10^{-27}$). This suggests that authors who shared data were more likely to include a data availability statement. Consequently, journals should require data availability statements in order to encourage more authors to share their raw data.

## Positive impact of journals policies on code sharing

In our study, we sought to investigate the differences in code availability policies across eight biomedical journals and examine the impact of these policies on code sharing. We found that out of the eight journals, five mandated code sharing, while three either encouraged code sharing or mandated it upon request (Table S1). In contrast, six of the eight journals had a mandatory data sharing policy, and two only encouraged or mandated data sharing upon request. Furthermore, across all journals, we see that 92.3% are sharing their code in non-notebook formats, R Markdown Document (RMD), or Notebook format (Fig. S10). To investigate the effect of journal policies on code and data sharing, we analyzed the relationship between the journal policy for code and data sharing and whether the actual code and data was available for that article for that specific journal.

For all 453 articles, articles were 2.3 times more likely to share code if the journal policy was mandatory rather than encouraged ($p = 1.9 \times 10^{-5}$). Furthermore, the number of articles whose journals mandated code and shared their code was significantly larger than the number of articles whose journals only encouraged sharing code and shared their code (Pearson's $\chi2 = 17.9$, $p = 2.4 \times 10^{-5}$). Moreover, when looking specifically at journals that only encouraged code sharing, we found that the percentage of code unavailable was much higher than the percentage of code available (Fig. S11).

Upon analyzing the percentage of articles sharing code across individual journals, we observed a notable contrast in Bioinformatics, and Nucleic Acids Research journals, where the difference between articles with shared code (Bioinformatics: 88.3%, Nucleic Acids Research: 18.6%) and those without (Bioinformatics: 11.7%, Nucleic Acids Research: 81.4%) respectively was significantly larger than in most other journals. The majority of other journals displayed approximately equal proportions of articles sharing code and

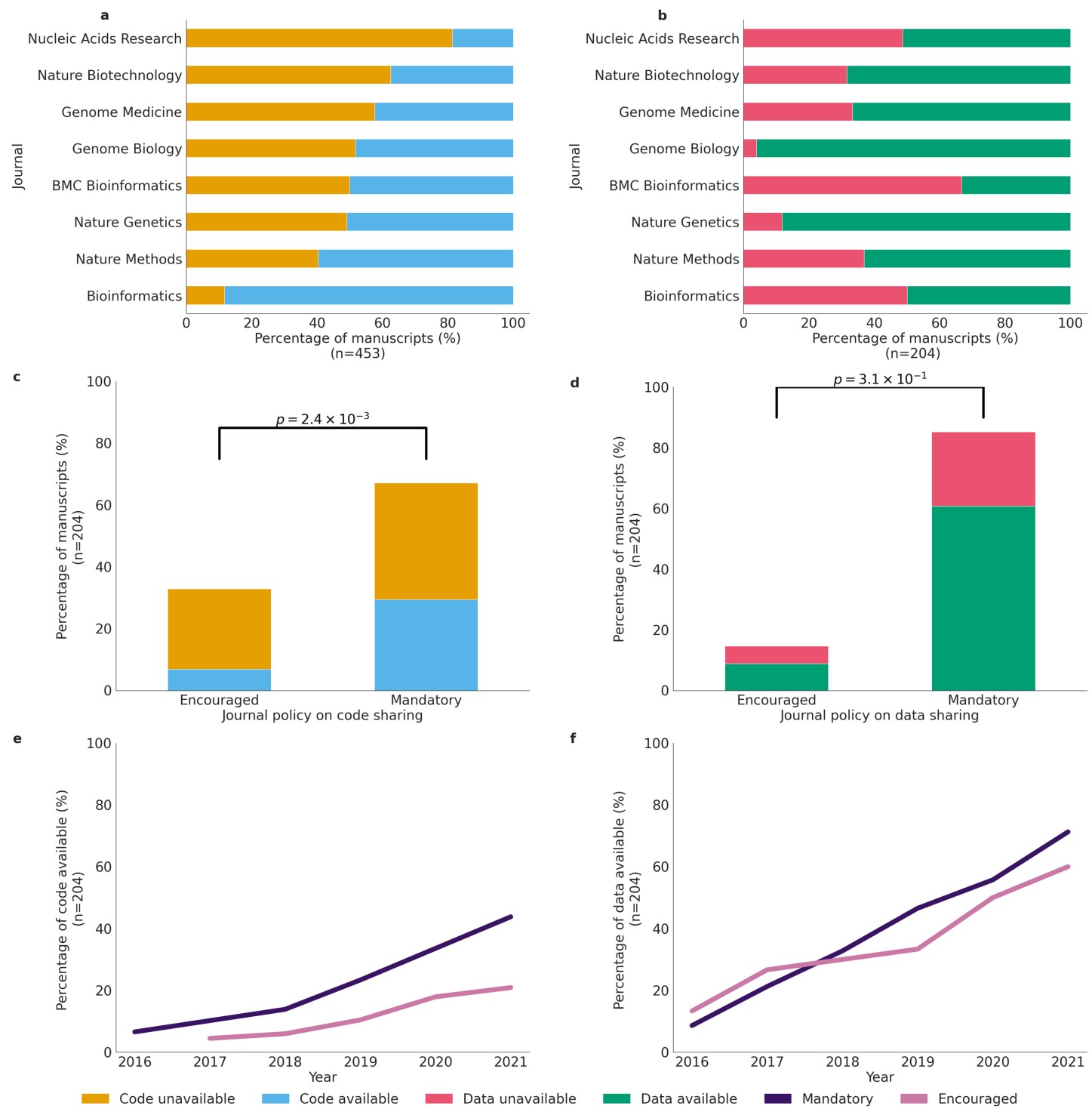

**Figure 3 Code and data availability across biomedical journals.** (A) Code sharing across all 453 studies per journal ($n = 453$). (B) Data sharing across all 453 studies per journal ($n = 204$). (C) Code sharing in primary analysis studies across eight journals, by journal policy ($n = 204$). (D) Data sharing in primary analysis studies across eight journals, by journal policy $n = (204)$. (E) Cumulative percentage of code availability for primary analysis manuscripts between 2016 and 2021 by journal policies (Mandatory, Encouraged) ($n = 204$). (F) Cumulative percentage of data availability for primary analysis manuscripts between 2016 and 2021 by journal policies (Mandatory, Encouraged) ($n = 204$).

articles not sharing code (Fig. 3A). Similarly, in examining the percentage of articles sharing data by individual journals, Nature Genetics and Genome Biology stood out with the most substantial difference between articles with shared data (Nature Genetics: 88.2%, Genome Biology: 96.0%) and those without (Nature Genetics: 11.8%, Genome Biology: 4.0%). Typically, the majority of journals featured a higher percentage of articles that shared data, as opposed to those that did not (Fig. 3B).

However, when we narrowed our analysis to only primary analysis articles, we found that articles were 2.9 times more likely to not share their code if the journal policy only encouraged code sharing ($p = 1.8 \times 10^{-3}$). Additionally, there was a higher percentage of code not shared across both types of journal policies (Pearson's $\chi2 = 9.2$, p = $2.4 \times 10^{-3}$). Nevertheless, the percentage of code available was much lower than the percentage of code unavailable for journals that only encouraged code sharing (Fig. 3C).

When looking at the effect of journal policies on data sharing, we found no significant effect despite the fact that journals that had mandatory data sharing policies had a much larger percentage of code shared than journals that only encouraged code sharing (Pearson's $\chi2 = 1.0$, $p = 0.3$). The lack of significance is most likely due to the limited sample size as out of all eight journals, only two journals had non-mandatory policies. However, in both categories, journals that had mandatory policies and journals that encouraged data sharing, the percentage of articles that shared data was higher than articles that did not share data (Fig. 3D).

Upon examining the 453 articles annually in accordance with journal policies, we observed a consistent positive trajectory in journals that mandated code sharing. The percentage steadily increased from 6.6% in 2016 to 43.8% in 2021. Journals that merely encouraged code sharing also exhibited a positive trend, albeit at a slower pace. The percentage of articles sharing code rose from 0.0% in 2016 to 20.9% in 2021 (Fig. 3E). Likewise, when scrutinizing the 453 articles yearly based on journal policies, we found that journals with a mandatory data sharing policy demonstrated a consistent positive trend in data sharing. The percentage of articles sharing data increased from 8.6% in 2016 to 71.3% in 2021. Articles from journals that only encouraged data sharing experienced an increase from 13.3% to 60.0% between 2016 and 2021 (Fig. 3F).

## DISCUSSION

Despite the growing recognition of transparency and reproducibility in biomedical research, there is no consensus on actionable steps for the scientific community when it comes to data and code sharing, elements that are essential for the advancement of science (*Stodden et al., 2016*). Our study, while based on a relatively small number of manuscripts ($n = 453$), which is approximately 3.5% of articles from the eight journals selected, provides a comprehensive analysis of code availability in biomedical research, highlighting the importance of sharing code and data. We propose that sharing code and data should be integral to study submission, similar to ethical review and publication of results. Journals and funding bodies should mandate code and data sharing as requirements for publication and funding. By doing so, they can inform and encourage the biomedical community to adopt effective strategies for improving transparency and reproducibility in data-driven

biomedical research. To address the obstacles hindering data and code availability, we aim to raise awareness and promote open dialogue among journals, funding agencies, and researchers.

One key benefit of making code freely available is its time-saving aspect, enabling researchers to adapt existing code and avoid starting analyses from scratch (*Barnes, 2010*). Additionally, code sharing fosters collaboration within and across research groups ("Data sharing and the future of science," *Nature Communications Editorial, 2018*; *Cadwallader & Hrynaszkiewicz, 2022*), leading to increased efficiency and novel discoveries. Moreover, code availability increases trust and enables researchers to reproduce and validate each other's results, enhancing the credibility of scientific research. Reusing code across various projects contributes to the development of standardized analytical tools and workflows, further improving research practices (*Marx, 2013*).

While data sharing became widely enforced, guidance on code sharing remains limited (*Goldacre, Morton & DeVito, 2019*; *Trisovic et al., 2022*). This imbalance highlights the need to develop comprehensive guidelines and standards that encourage researchers to share code along with their research results (*Trisovic et al., 2022*). Practices promoting the open sharing of code products and protocols significantly contribute to reproducibility, facilitate secondary analysis, and prevent unnecessary barriers in republishing previous findings (*Trisovic et al., 2022*).

To foster this collaborative and efficient research environment, the scientific community must promote open science practices, establish clear standards for code sharing and citation, and develop guidelines for reporting preclinical research data (*Goldacre, Morton & DeVito, 2019*; *Trisovic et al., 2022*; *Eglen et al., 2017*; "Enhancing Reproducibility through Rigor and Transparency" *NIH*; *Wilkinson et al., 2016*). However, studies indicate that supporting data reporting in research publications remains inadequate (*Goldacre, Morton & DeVito, 2019*; *Kilkenny et al., 2010*). For instance, only 38.1% of immunogenomics research publications shared their data on public repositories (*Huang et al., 2022*). Therefore, it is important to develop and adopt specific guidelines for sharing code products used in analysis and visualizations, similar to those for data sharing (*Trisovic et al., 2022*). Even so, the debate on how scientists should go about sharing code remains unresolved.

Typically, researchers will share their code using open-source repositories like GitHub. However, relying on these platforms poses challenges, with GitHub being a short-term storage solution, with limited file capacity, concerns about archival stability and unreliability due to unstable URLs over time. Since a GitHub owner has the ability to delete their code at any time, it is important to note that archiving code in GitHub does not make the code stable or open source. For stability, GitHub should be paired with another platform that enables minting of stable DOIs (Box 1); for example, the EU Open Research Repository (*European Commission, 2013*), Bioconductor (*Gentleman et al., 2004*), or Software Heritage (*Di Cosmo, 2020*), to name a few. These platforms allow for tracking changes over time, and it is important to follow the Software Citation Guidelines when writing a manuscript to clearly state which version of the software was used in the study and to cite the DOI of the software itself (*Katz et al., 2020*). To make code open source, an

appropriate OSI license should be used. Nonetheless, the scientific community must engage in a dialogue about the long-term implications and explore alternative solutions to ensure the accessibility and stability of shared code and data in biomedical research.

---

**Box 1.** Actions to increase availability and archival stability of analytical code. Here, we present four principles to increase code availability and archival stability.

**1. Software and resources should be hosted on archivally stable services**

Authors should avoid putting code into Supplemental Files as Supplemental Files are often not seen or "lost" at the end of a manuscript. All code should be added into a stable repository along with clearly labeled Supplemental Files, and documentation. Selecting the appropriate service to host your software and resources is critical. A simple solution is to use web services designed to host source code (*e.g.*, GitHub, SourceForge, Software Heritage). For GitHub users, it is important to note that to ensure long-term stability of code, GitHub should be paired with other stable platforms (*e.g.*, Zenodo or Software Heritage), which enables users to easily take their own snapshots of their code and have citable DOIs.

**2. Streamlining Installation with Intuitive Interfaces and Comprehensive Scripts**

To facilitate simple installation, provide an easy-to-use interface to download and install all required dependencies. Ideally, all necessary installation instructions should be included in a single script, especially when the number of installation commands is large. Package managers such as pip or anaconda can potentially make this problem easier to solve.

**3. Journals should incentivize authors to make code publicly available.**

Journals should incentivize authors to make code publicly available. Code and data sharing should be considered the main part of submitting a study, similar to how we consider ethical review and publication of study results. Journals and funding bodies should insist that authors make raw data and other supporting data available. Moreover, they should follow recommendations of the National Academies regarding data sharing and abide by the guidelines of the Fort Lauderdale and Toronto agreements (*National Research Council, 2003*). Journals should mandate clear source code and data availability sections to be included in all submitted manuscripts, where accessions for raw data in community-approved repositories are made clear, and other data supporting is cited. Furthermore, reviewers should be asked to check and test code where possible. Other supporting data should be made openly available in a public server like Zenodo or Figshare.

To further underscore the importance of code and data sharing, the evolution towards making it an integral aspect of the submission process is gaining momentum. Ideally, journals will eventually require such sharing upon submission rather than post-acceptance. This paradigm shift from considering it merely as good research practice to deeming it a necessary policy change is essential for fostering transparency and reproducibility. Enforcing code and data sharing at the submission stage empowers reviewers to scrutinize the study's code, potentially identifying and rectifying any

*(Continued)*

(continued)

methodological errors early on. This proactive approach not only aligns with the broader ethos of open science but also addresses the prevalent issue of retractions, where errors in code and data have played a significant role (*Heyard & Held, 2022*). Authors must enter a commitment that their code and data will have perpetual access in stable software platforms such as Zenodo, Software Heritage, Bioinformatics.org. Snapshots of code taken with citables DOIs should be considered a sign of commitment for the long term. And to ensure extremely long-term preservation, several digital preservation services, such as CLOCKSS (*Clockss, 2023*) and (*Portico, 2022*) are already being used by some publishers, including Code Ocean who has partnered with CLOCKSS to ensure long-term preservation of code (*Staubitz et al., 2016*).

**4. Training, seminars and conferences should be held on code and data sharing** Holding seminars and conferences about the advantages of sharing data and code, including hands-on workshops, such as Software Carpentry. PLOS Computational Biology journal presented a new code-sharing policy in March 2021. This policy focuses on improving code sharing. This policy increased repeatability of studies after 2021.

Sharing research data and code is fundamental for advancing scientific knowledge, enabling collaboration, and accelerating progress. It prevents duplication, allows building upon existing work, and generates critical new findings in the biomedical field and other domains. Openly sharing research data and code ensures the reusability of studies, promotes transparency, and enhances the credibility of research findings (*Goldacre, Morton & DeVito, 2019*; *Trisovic et al., 2022*; *Huang et al., 2022*; *Brito et al., 2020*). Several recommendations and agreements are already in place for sharing data, such as the Fort Lauderdale agreement in 2003 that reaffirmed the 1996 Bermuda Principles (*The Wellcome Trust, 2003*), the Toronto agreement in 2009 ("Prepublication data sharing," *Toronto International Data Release Workshop Authors, 2009*), and data sharing recommendations by the National Academies (*National Research Council, 2003*). Complementing these efforts, there exists hands-on education initiatives, such as Software Carpentry, which teaches fundamental analytical skills for research computing (*Software Carpentry, 2024*; *National Research Council, 2003*). However, there is no single agency responsible for enforcing compliance with these guidelines and procedures. This is a critical oversight, as it means that there is no one entity ensuring the usability and reproducibility of shared data and code. Therefore, it is crucial to identify a responsible agency to fill this role. The community should work towards establishing actionable strategies that align with the Findable, Accessible, Interoperable and Reusable (FAIR) guiding principles (*Wilkinson et al., 2016*; *Schulz, 2018*) to accelerate the sharing of research products. By merging efforts to enhance transparency and reproducibility in biomedical research, we can overcome these challenges and pave the way for more robust and reliable scientific findings.

## Feasible actions to increase availability of analytical code

The push for greater transparency and reproducibility in research practices gained momentum when journals implemented policies to ensure code and data availability, unless ethical or legal restrictions prevented it. As a result of these rule changes, an increasing number of researchers began adhering to these rules. (*Cadwallader et al., 2021*). Furthermore, our analysis suggests that journal policies can have a significant effect on the availability of code. Journals that require code are more likely to publish articles with code available. Notably, an analysis of studies utilizing natural language processing unveiled a rising trend in code-sharing compliance. The introduction of a new policy mandating code sharing resulted in a rate of 53.0% in 2019, 61.0% in 2020, and 73.0% in 2021. Subsequent to the implementation of this policy, the code-sharing rate further surged to 87.0% in 2022 (*Cadwallader et al., 2022*).

A noteworthy example illustrating this shift in policies is the initiative undertaken by Nature journal. At the forefront of this initiative is the option for authors to formally "opt in" to collaborate with platforms such as Figshare or Code Ocean during the submission process. This innovative feature empowers journal teams to efficiently verify the presence of requisite links, ensuring accessibility, and confirming the review status of associated data and code. Promoting the adoption of these options among researchers is highly advisable. At the present juncture, this integration is exclusively available within the domains of Nature Computational Science and Nature Machine Intelligence. However, it is reasonable to anticipate a broader rollout of this integration across additional journals in the near future ("Seamless sharing and peer review of code," *Nature Computational Science Editorial, 2022*). Regrettably, not all journals maintain consistency in regard to code and data sharing, and each journal or publisher establishes its distinct policies concerning source code and data availability. While certain journals, like GigaScience and GigaByte, adopt a more open approach, others like Nature simply offer authors the choice of embracing openness without making full transparency mandatory. Efforts towards achieving reproducibility demand substantial commitment from authors, editors, reviewers, and publishers. Despite some publishers providing authors with the option to pursue an "open/reproducible" approach, authors often tend to choose the path of least resistance unless such practices are mandated. Unfortunately, this choice often falls short of the ideals associated with open science and reproducibility.

Furthermore, there are two main ambiguities in the policies of some journals. First, it is unclear whether "supporting data" and "underlying data used for analysis" refer only to raw data or also include software source code, algorithms, and code used for analysis. Second, it is unclear whether the policy includes making all code used for analytics available or only software source code or custom code. Efforts have been made by the biomedical community to establish effective strategies and best practices to enhance reproducibility and rigor in research such as Code Ocean, a cloud based reproducibility platform that has been implemented by a number of journals including, GigaScience since

2017 (*Edmunds, 2017*), and since 2020, several Nature journals, BMC Bioinformatics, Scientific Data, and Genome Biology (*Cheifet, 2021*). Sharing code and ensuring reproducibility has extended beyond the non-biomedical publishing space; the Political Analysis journal wraps every articles' methods in Code Ocean as standard ("Political Analysis" *Cambridge Core*). Since 2012, the pioneering open science journal, GigaScience, has been mandating all articles to have open data, supporting metadata and source code with Open Source Initiative (OSI)— approved licenses ("Licenses" *Open source initiative, 2022*). The journal also has its own repository, GigaDB (*GigaDB, 2011*), where authors can host data (not already open in a community approved repository, *e.g.*, NCBI), in the public domain under a CC0 license. To ensure reproducibility and stability of source code, the GigaDB Curators also take snapshots of all code and a dataset is produced with a citable Datacite DOI (Digital Object Identifier). There are new publishing efforts that aim to address transparency and reproducibility in research by moving beyond static publications. One new journal, GigaByte (*Gigabyte, 2020*) publishes articles with embeddable and interactive features, such as 3D-image viewer *via* SketchFab, interactive maps, videos, Hi-C data figures, and methods *via* Protocols.io. In addition to publishing the epitopepredict (*Farrell, 2021*), GigaByte also published a Stenci.la (*Stencila : Dynamic documents made simple, 2018*) Executable Research Article that enables readers to see and play with the code underlying the supporting figures. The epitopepredict tool was also wrapped in a Code Ocean capsule and embedded in the online article, allowing readers to interact and test the code, as well as deploy it to their own AWS Cloud compute service. Furthermore, GigaByte provides a checklist for reviewers to go through which includes checking data availability and source code ("Gigabyte"). eLife has also leveraged the Stenci.la platform as part of their Reproducible Document Stack initiative ("Reproducible Document Stack—supporting the next-generation research article," *eLife, 2017*).

This is all part of a larger effort to improve the reproducibility of scientific research. CODECHECK is an open science initiative that tackles one of the main challenges of reproducibility by providing guidance, workflows and tools that enables anyone to independently test the execution of code underlying research articles (*Eglen & Nüst, 2019*). Some journals have already adopted this scheme, where a citable, time-stamped CODECHECK certificate is issued, and credit also given to the Codecheckers (*Nüst & Eglen, 2021*). CODECHECK is an easy scheme any publisher and journal can use to ensure reproducibility of articles, and is also a great learning tool for students or anyone interested in data science. Nevertheless, an ongoing dialogue between journal publishers and researchers is necessary to overcome perceived and technical barriers to transparent and reproducible data-driven research across various biomedical disciplines.

Data-driven biomedical research relies heavily on computational analyses, yet there is currently a lack of effective mechanisms to ensure transparency and reproducibility of data analyses performed to generate results in scientific publications. Despite efforts to increase transparency and reproducibility, many published studies remain non-reproducible due to the lack of code and data availability (*Trisovic et al., 2021*). One reason for this is the researchers' concerns about losing priority over their novel results, misrepresentation of their work, maintaining exclusivity of their data for future projects, or a fear of data

parasitism (_Park & Greene, 2018_), as well as lack of credit for making data and source code open. These concerns can be addressed by including the primary authors as co-authors on any publications resulting from previously shared code and data (_Elliott & Resnik, 2019_), as well as citing software correctly by citing the article describing the software and the source code (_Katz et al., 2020_).

To address all of these challenges, code and data sharing should be considered an integral part of study submission, similar to ethical review and publication of study results. They should also be considered as early as possible when developing research projects, and sharing policies for studies should be conceived as part of the study design (Box 1). The neuroimaging community are strong adopters of open science and have suggested four important aspects of scientific research practice: respect trademarks, clarify ownership by looking at copyright status, release code and data under free and open licenses, and finally obtain the permission to share the data or code (_Halchenko & Hanke, 2015_). In addition, Stanford Data Science have released their Stanford Open By Design Handbook, targetted to early career researchers who want to adopt open science practice, but lack the knowledge and expertise ("Stanford Open By Design" _Stanford Data Science, 2022_).

Journals and funding bodies should require authors to make raw data available by publishing it on public servers and committing to long-term availability of their code and data. Universities should also provide open science guidelines and support for their researchers. By implementing such practices, researchers can enhance transparency, reproducibility, and the overall quality of data-driven research in the biomedical field.

### Funding
Serghei Mangul, Nitesh Sharma, Ram Ayyala, Dhrithi Deshpande, and Mohammad Vahed were supported by the National Science Foundation (NSF) grants 2041984, 2135954 and 2316223 and National Institutes of Health (NIH) grant R01AI173172. The funders had no role in study design, data collection and analysis, decision to publish, or preparation of the manuscript.

### Grant Disclosures
The following grant information was disclosed by the authors:
National Science Foundation (NSF): 2041984, 2135954 and 2316223.
National Institutes of Health (NIH): R01AI173172.

### Competing Interests
Nicole Nogoy is an Executive Editor for GigaScience Press.

### Author Contributions
- Nitesh Kumar Sharma conceived and designed the experiments, performed the experiments, analyzed the data, performed the computation work, authored or reviewed drafts of the article, and approved the final draft.

- Ram Ayyala conceived and designed the experiments, analyzed the data, performed the computation work, prepared figures and/or tables, authored or reviewed drafts of the article, and approved the final draft.
- Dhrithi Deshpande conceived and designed the experiments, performed the experiments, analyzed the data, authored or reviewed drafts of the article, and approved the final draft.
- Yesha Patel conceived and designed the experiments, performed the experiments, authored or reviewed drafts of the article, and approved the final draft.
- Viorel Munteanu conceived and designed the experiments, performed the experiments, authored or reviewed drafts of the article, and approved the final draft.
- Dumitru Ciorba conceived and designed the experiments, performed the experiments, authored or reviewed drafts of the article, and approved the final draft.
- Viorel Bostan conceived and designed the experiments, authored or reviewed drafts of the article, and approved the final draft.
- Andrada Fiscutean conceived and designed the experiments, performed the experiments, authored or reviewed drafts of the article, and approved the final draft.
- Mohammad Vahed conceived and designed the experiments, performed the experiments, authored or reviewed drafts of the article, and approved the final draft.
- Aditya Sarkar conceived and designed the experiments, performed the experiments, authored or reviewed drafts of the article, and approved the final draft.
- Ruiwei Guo conceived and designed the experiments, performed the experiments, authored or reviewed drafts of the article, and approved the final draft.
- Andrew Moore conceived and designed the experiments, performed the experiments, authored or reviewed drafts of the article, and approved the final draft.
- Nicholas Darci-Maher conceived and designed the experiments, performed the experiments, authored or reviewed drafts of the article, and approved the final draft.
- Nicole Nogoy conceived and designed the experiments, authored or reviewed drafts of the article, this author supervised this work, and approved the final draft.
- Malak Abedalthagafi conceived and designed the experiments, authored or reviewed drafts of the article, this author supervised this work, and approved the final draft.
- Serghei Mangul conceived and designed the experiments, performed the experiments, authored or reviewed drafts of the article, this author supervised this work, and approved the final draft.

## Data Availability

The data and code is available in the Supplemental Files and at GitHub and Zenodo:
- https://github.com/Mangul-Lab-USC/code-availability.
- Ram Ayyala. (2023). Mangul-Lab-USC/code-availability: V1 (test). Zenodo. https://doi.org/10.5281/zenodo.10108745.

## Supplemental Information

Supplemental information for this article can be found online at http://dx.doi.org/10.7717/peerj-cs.2066#supplemental-information.

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
