# Peer review of "Analytical code sharing practices in biomedical research"

_PeerJ Computer Science, doi:10.7717/peerj-cs.2066_

## Round 0.1 · original submission · Minor Revisions

Thanks for this submission. The reviewers found your paper to be interesting and worthwhile. Although the suggested revisions are minor, they are important. Please respond to each of the suggestions below.

·

Basic reporting

The article formatting is lacking. Citations are missing spaces or are not consistent in formatting, numbers (statistics) are formatted inconsistently, and figure captions are partially or completely missing. Some sentence and paragraph structure is odd or repetitive. I think 10-20% of the text could be trimmed out where it repeats something that has already been stated. Other than this, the article is mostly well-written and understandable.

Experimental design

The methods could use slight improvement of clarity (see additional comments).

Validity of the findings

Data have been provided, Code in the form of a python notebook has been provided. Statistics are simple as they should be for these simple comparisons.

Additional comments

Overall, I like this paper. I think we need more analyses like this to assess the state of open data and open code across different fields and journals. I’d like to see more of the supplementary figures make it into the main body of the manuscript. Most people don’t end up seeing the supplementary figures. I also think the authors need to better highlight that GitHub is not a viable long-term repository option. Code and data must be given a DOI, which GitHub does not do. GitHub does have integration with Zenodo, but the authors need to make this message explicit so readers don’t think that publishing code or data via GitHub is an acceptable option, because it is not. Other than this, I have mostly minor comments. For example, some of the formatting contains errors.

Dylan Gomes
* * *
In abstract, there are a few long run-on sentences that could be improved. These sentences would be clearer if broken up into multiple sentences.

There are some p values in the abstract and throughout the manuscript, one is represented with an “x” as a multiplication sign and another with an asterisk “*”. Asterisks are commonly used as multiplication in programming languages, but not typically in the written English language. These should, at the very least, be consistent, but I would recommend removing the use of the asterisk throughout as a multiplication sign.

The authors also represent 10-7 as “10-07”, which isn’t a common practice and should be replaced (throughout the manuscript) with the superscript to denote the power function.
This point may be moot for the abstract, however, as I would recommend removing the p values from the abstract. Summary statistics are too much detail for an abstract, in my opinion.

Lines 65-67: This sentence is oddly worded and has some formatting issues.
Throughout there are many citations that have formatting issues (e.g., no spaces, etc.)

The last paragraph of the introduction doesn’t belong in an introduction. It includes results and discussion material and reads similarly to the abstract.

Were the 12,603 articles ALL of the articles published in those 8 journals during that time? It isn’t clear from the text in the methods that this is the case. While this isn’t exactly a systematic review, it might be helpful to have a small PRISMA diagram.

While 453 manuscripts is a substantial amount of work, it is worth noting that this is only about 3.5% of the articles from those journals. Randomly sampled or not, the authors should acknowledge that this may not be a representative sample.

I’d really like to see a figure of the code sharing % across the different 8 journals – to call the journals out specifically.

Table 1 very nicely shows which journals have “mandatory” code policies – are they ACTUALLY mandatory? This could be also displayed by Mandatory vs Encouraged – does the code policy make a difference in who shares code? This could be very helpful for journal editorial teams / policy makers. Ahh, I see that these figures are in the supplemental materials. I strongly encourage authors to add additional figures to the main text. These figures tell a compelling story and are nicely made, why not showcase them in the manuscript? I don’t believe PeerJ has space limitations.

Also, there are no captions for the figures in the supplement, so it is difficult to know what is going on with them.

For the data and code sharing platforms, I think a very important figure would be to demonstrate which proportion of platforms are not good long-term storage options. GitHub, for example, does not provide DOIs for data or code, and thus it is not a viable long-term solution for data or code archiving, unless paired with another platform (e.g., Zenodo) that allows minting of a DOI. In other words, anyone can delete their GitHub account and every dataset and code “published” there will be lost forever. I think this would be worth discussing in your manuscript. This is briefly mentioned in the Gomes et al. 2022 reference you mention in the Introduction if you are looking for more information. In the results you mention that only ~6% is published in Zenodo! This is an important headline – even those that share code are not doing so in long-term repositories!! I think this is important enough to be highlighted in the abstract. Line 213-214 starts to get at this long-term stability, but this 1% number is the minimum number of links that won’t work in the future. The further we get from the publication date of those other papers, the fewer the links that will work.

The GitHub link mentioned in the manuscript for the data and code is not live or working, but as I mentioned before, this is not a long-term option. Via GitHub, however, you can permanently archive with Zenodo as Zenodo is integrated into GitHub. In my opinion having a DOI is a requirement for actually sharing data and code, otherwise there is nothing ensuring it’s longevity.
The Figure 1 caption is cut off such that I don’t know what Figure 1 d-f is showing.

Line 267-268 does not make sense to me. You are saying that the data availability statement is making the researchers share data? This logic seems backwards. There is an association because if authors share data they are more likely to add a data availability statement, not the other way around. You haven’t assessed how often these datasets are accessed, right?

Line 281-284: this sentence also doesn’t make sense to me. You say people are 0.43 times more likely to share code with mandatory rules but 2.34 times more likely not to if the journal policy is not mandatory. Aren’t these the same data? The number should be the same, right? Furthermore, 0.43 times more likely is somewhat confusing. A factor of 0.43 would be a reduction, so it must be a factor of 1.43, to make it 0.43 times more likely (an increase in 43%), is that right? An increased probability of 43% is more digestible to readers I think.

Lines 281-289: this entire paragraph appears to repeat the same information 3 times in 3 different sentences. If these are meant to convey different information, perhaps you can clarify this. Otherwise, you could significantly cut the amount of text you are using here – and throughout the manuscript. There is a lot of repetitive text.

Lines 298-303: Do you think this was not significant because there was only two journals with non-mandatory policies, which limited the sample size of that group, giving it high error?

Line 309: Everything following this header is Discussion material, not Results.
The first paragraph of this section is very long – you might consider breaking it up to ease the reader’s eyes.

Line 427: GitHub is not open-source. It offers free software and repository hosting, but is not open-source, counter to what you suggest here. You use open-source again in the next sentence when I think you just mean “open” not “open-source”.

Lines 429-431: I disagree that we should encourage this. Instead just include something along the lines of researchers can easily mint DOIs for their data/code through GitHub’s integration with Zenodo… Or via OSF or others. Box 1 should be updated to reflect this. GitHub is NOT a long-term data or code storage option (by itself).

Cite this review as

Reviewer 2 ·

Basic reporting

This is a very interesting paper and I think is a good fit for PeerJ Comp Science. I believe that the analysis is clearly reported, although I list below some suggestions for clarifications. In my opinion, these are optional and can be added/discussed at the authors' discretion.

Experimental design

The selection of 10 random articles over 5 years may not be representative of the trends of each journal. How feasible would it be to sample more articles?

This study is novel, in my opinion. I haven't been aware of anyone who has tried to quantify these trends, although I do think that the idea that more data/code is being shared in recent years compared to 5 years ago is well-known. It also isn't surprising that when journals require data/code sharing, more code and data are shared, but it is interesting to see the data on this.

Please note that I am not qualified to comment on the statistics.

Validity of the findings

Some questions:

Should code included in supplemental files be also deposited on a repository?

For papers that had a Code Availability statement but did not share any code - is this because there was no code present in the paper (line 221)?

You mention that a proportion of code that is said to be available in a publication is not actually available (due to unstable links). Is there a solution for what can be done about these?
Similarly, you mention in Box 1 that authors must make a commitment to have their code stable for a "long time" - what would be the cutoff here? Until something better comes along? Until it stops getting accessed by the community? I should note that Code Ocean also has partnered with CLOCKSS (https://codeocean.com/preservation/) to address this problem.

Do all the journals that have code sharing requirements also have dedicated Code Availability and Data Availability statements? It seems like it would be a good idea to have these statements clearly made in all journals - otherwise how will the community easily find data/code.

I would also add that it would be useful for journals to not only require code/data sharing, but to make sure that data and code is reviewed.

Another thing to note is that journals are trying to better integrate code and data sharing with submissions: https://www.nature.com/articles/s43588-022-00388-w
The idea is that you can "opt in" to working with figshare or Code Ocean at submission stage, and journal teams can easily check to make sure that links are included and data/code is available and reviewed. It would be good to encourage researchers to take up these options. Right now this integration is only at Nature Computational Science and Nature Machine Intelligence but it would be reasonable to expect a greater roll out of this.

Do you think it is necessary for all code to be reported? Should journals be checking for code used to make figures, for example (reproducibility), or does that sort of custom code not need to be deposited or reported? For the papers that you've looked at - did you check to see if all the code is reported, or if just the major method/code is?

You mention data deposition - what about synthetic datasets? Should they be included with the code deposition or as separate data?

It would be interesting to have a breakdown of the stats/percentages per journal as a separate table. Although that data seems to be included in more detail in the raw file - it would be more clear to see the trends between Table 1's details and the percentages of actual code/data availability at each.

Additional comments

Some of the final section of the results should possibly be moved into the discussion section.

Also, the introduction talks about retracted papers - I think this is opening a can of worms unless the authors give a bit more context. Yes, more papers are retracted in recent years, but there are also a large number of papers being published, much more data available, and more people who are looking out for this sort of thing. Some more context would be needed here to say how this actually relates to code/data sharing. It could actually be said that more data/code sharing leads to more retractions - because more people have access to the data to reuse or check it. So I'm not sure that presenting retractions here in the negative light is the best way to talk about this trend. Also, retractions are due to a whole host of reasons.

Cite this review as

·

Basic reporting

The paper overall is clear and easy to follow. Most of the literature is relevant.

I do have some comments though.

line 61 - literature on retractions is from 2011; I would suggest searching for more recent review. e.g. from 2022 https://pubmed.ncbi.nlm.nih.gov/34921899/ but there are probably better references.

line 206: reference to Table 1 seems wrong. The Table does not show these numbers, but simply the journals.

Why did you select the journals that you did? Could you provide a description of your criteria. It might have been nice to select PLOS Comp bio, following their introduction of a code sharing policy (see lines 310 onward), but I'm not suggesting at this stage that journal be included.

l349: our codecheck project has a paper which might be a better (long-term, stable) object to cite than our website: https://doi.org/10.12688/f1000research.51738.2. (noting the obvious declaration of interest)

Experimental design

Why did you select the journals that you did in Table 1?

l122: where is the python script that was used?


l131: it is not clear whether you wrote programs to analyse the papers, or whether you went through them manually to search for the information that you were looking for. If the former, are the programs available? If the latter, how long did it take you, on average, to scan one paper, given that you had 450 papers to read? (That would be a lot of work.)

Either way, it would be useful to know how to repeat this analysis, e.g. to update the paper at some point for 2022 and future years.

Validity of the findings

The findings appear appropriate, and show an increasing (encouraging) trend to increase availability.

Additional comments

I'm not clear why the phrase 'analytical code' is used rather than e.g. 'research code' or just 'code'. Analytical code might rule out in my mind other types of code, such as 'modelling code', since code is written for modelling as well as analysis.

The citations throughout the article need a space before an opening parenthesis.

From line 300 onwards, this material might be better placed in the discussion than the results, as it is no longer reporting on the findings from the paper.

line 427: an obvious remedy for the long-term repositories is Zenodo (or other long-term storages), and there are integrations with GitHub that provide automated archival of releases from Github.

line 454: box 1 -- as well as showing actions how to share code, one extra recommendation they might wish to consider is when to share the code. This article advocates for sharing upon submission, which I think is an interesting (but not always applicable) approach.

https://doi.org/10.1111/1740-9713.01623. (Heyard and Held, 2022).


Figure 1 is quite grainy, and appears to be a low-resolution bitmap. I had trouble reading the axes.

Figure 1f - please clarify if these figures are cumulative, or just year-by-year figures. (I don't think it makes sense to show cumulative data here.)

The legend of the figure in my pdf provided by the journal (page 22) was truncated during panel C.

The article comes with 11 supplementary figures, and only one main figure. Could you consider moving them (all?) into the main article, so that it is easier for a reader to follow the argument. My rule of thumb is to minimise supplementary figures, especially when there are no space constraints from a print journal.



I was not able to re-run the Figures.ipynb. I hit a few problems.

First, when working with the files provided by the journal, the
notebook errored as the data file

../data/240_B.txt

was not provided.
* * *
However, if I cloned the repository:
https://github.com/Mangul-Lab-USC/code-availability

I could then get a bit further. I needed to install several packages.
The most problematic was meeting the following dependency:

from pysankey import sankey

I installed pysankey using

pip install --user pysankey

but this still didn't satisfy the dependency. If I changed the 'from'
line to read:

from pySankey import sankey

the notebook worked up until the code chunk for figure 1b, which
generated:

TypeError Traceback (most recent call last)
Cell In[11], line 16
13 for i, label in enumerate(category_labels):
14 color_dict[label] = colors[i]
---> 16 sankey(
17 left=df_temp['Code availability statement (Yes/No)'].values, right=df_temp['Code availability (Yes/No)'].values, rightWeight=df_temp['Counts'].values.astype(float), leftWeight=df_temp['Counts'].values.astype(float), aspect=20,
18 fontsize=10,colorDict=color_dict
19 )

TypeError: 'module' object is not callable
<Figure size 800x1000 with 0 Axes>

Perhaps I installed the wrong package for pysankey? Could you provide
a requirements.txt file that specifies what packages (and their
versions) are required? This would seem to be good practice for
reproducibility. See https://learnpython.com/blog/python-requirements-file/


https://doi.org/10.1111/1740-9713.01623

The authors might wish to provide a notebook that runs remotely on google colab or mybinder so that these dependency issues can be resolved, and anyone can run the code without additional package installations.

Cite this review as

---

## Round 0.2 · Minor Revisions

Thank you for your extensive response to the first round of comments. Reviewers agreed that this paper is significantly improved and should be published. In your final subnmission, please be sure to address the outstanding reviewer comments regarding figure 1f and the provided code notebook. These comments should be easy to address.

Reviewer 2 ·

Basic reporting

No additional comments

Experimental design

No additional comments

Validity of the findings

No additional comments

Additional comments

Thank you for thoroughly addressing my points in the revised manuscript. I have no more comments to add.

Cite this review as

·

Basic reporting

Thank you for the detailed rebuttal letter and for the updates. I am happy with the changes that you have made. I just have two final comments.

1. Figure 1f:

You state "Figure 1f is cumulative as described in the Figure caption. We plotted cumulative increases in code sharing to show how each year, the total number of manuscripts out of the 453 are sharing code, to show that the trend of sharing code is increasing year to year. ".
I'm still struggling to understand this figure though. e.g. for the 2016 data, it shows 7.3% papers. but what is the raw count of papers published in the year? Is there a total of 453 papers over the years 2016-2021? The y axis tells me n=453, but without knowing the number of papers published in 2016, I cannot work out the percentage. Can you start by showing the raw data -- how many papers per year had code/data available?

2. Notebook

Thank you for providing the requirements.txt file; this helped me get a bit further. I did however hit another problem with the code; Figure 2 would not regenerate, but instead I got this error in the notebook:

NameError Traceback (most recent call last)
Cell In[24], line 7
4 rcParams['font.family'] = 'sans-serif'
5 sns.set_context("paper",font_scale=1)
----> 7 ax=sns.barplot(data=df_temp, y='Source', x='Percentage',hue='Code Sharing Link Expired?',errorbar=None,palette=color,legend=None)
8 sns.despine()
9 ax.set_xlabel("Percentage of code sharing links (%)\n(n=453)",fontsize=fontsize,loc='center')

NameError: name 'color' is not defined

Experimental design

no comment

Validity of the findings

no comment.

Additional comments

none.

Cite this review as

---

## Round 0.3 · accepted · Accept

Thanks for your thoughtful responses to the comments from reviewers.